# The short and long-term impact of COVID-19 restrictions on women's mental health in Mwanza, Tanzania: A longitudinal study

Heidi Stöckl[1,2]*, Neema Mosha[2,3], Elizabeth Dartnall[4], Philip Ayieko[3,5], Grace Mtolela[3], Gerry Mshana[3,6]

1 Department of Global Health and Development, London School of Hygiene and Tropical Medicine, London, United Kingdom, 2 Institute for Medical Information Processing, Biometry and Epidemiology, Ludwig-Maximilians-Universität München, Munich, Germany, 3 Mwanza Intervention Trials Unit, Mwanza, Tanzania, 4 Sexual Violence Research Initiative, Pretoria, South Africa, 5 Faculty of Epidemiology and Population Health, London School of Hygiene and Tropical Medicine, London, United Kingdom, 6 National Institute for Medical Research, Mwanza, Tanzania

* heidi.stoeckl@lshtm.ac.uk

**Data Availability Statement:** Participants in the current study have not given approval to share their anonymized data beyond the original authors. To access the data, please contact the authors for

## Abstract

The COVID-19 outbreak had a profound impact on all countries in the world, leading governments to impose various forms of restrictions on social interactions and mobility, including complete lockdowns. While the impact of lockdowns on the emerging mental health crisis has been documented in high income countries, little is known whether and how the COVID-19 pandemic also effected mental health in settings with few or no COVID-19 restrictions in place. Our study therefore aimed to explore the impact of few and no COVID19 restrictions on the self-reported mental health of women in Mwanza, Tanzania. The longitudinal study integrated a nested phone survey with two time points into an existing longitudinal study in Mwanza, Tanzania. In total, 415 women who were part of an existing longitudinal study utilizing face-to-face interviews participated in both phone interviews, one conducted during COVID-19 restrictions and once after the restrictions had been lifted about the prior three months of their lives. They also participated in a face-to-face interview for the original longitudinal study three months later. Using a random effects model to assess changes in symptoms of poor mental health, measured through the SRQ20, we found a significant difference between the time during COVID-19 restrictions (20%) and after COVID-19 restrictions were lifted (15%), and after life resumed to pre-COVID-19 times (11%). Covid-19 related factors associated with poor symptoms of mental health during restrictions and after restrictions were lifted related to COVID-19 knowledge, behaviour change, economic livelihoods challenges, increased quarrels and intimate partner violence with partners and stress due to childcare issues. Despite Tanzania only imposing low levels of restrictions, the COVID-19 pandemic still led to an increase in women's reports of symptoms of poor mental health in this study, albeit not as pronounced as in settings with strict restrictions or lockdown. Governments need to be aware that even if no or low levels of restrictions are chosen, adequate support needs to be given to the population to avoid increased anxiety and challenges to economic livelihoods. In particular, attention needs to be given to the triple burden that

joint analysis. Alternatively, for approval to access the data for this study without participation of the original authors, please write to the Medical Research Coordinating Committee, National Institute for Medical research, P.O. Box 6953, Dar es Salaam, Tanzania, ethics@nimr.or.tz, referencing IRB Approval No. NIMR/HQ/R.8c/Vol.I/614 and NIMR/HQ/R.8c/Vol. I/837.

**Funding:** HS received an ERC Starting Grant (716458) from the European Research Council. https://erc.europa.eu/homepage The funder played no role in the study. HS and GM received an NBER Gender in the Economy Study Group Research Grants on Women, Victimization, and COVID-19. National Bureau of Economic Research. www.nber.org. The funder played no role in the study.

**Competing interests:** The authors have declared that no competing interests exist.

women face in respect to reduced income generating activities, relationship pressures and increased childcaring responsibilities.

## Introduction

The COVID-19 outbreak was sudden and unexpected in most countries. The first known cases occurred in December 2019, and on March 11 in 2020 the World Health Organization declared it a pandemic. COVID-19 not only came with a high death toll and long-term consequences for many people, it was also accompanied by a wave of hidden pandemics caused by the unprecedented containment strategies to rein in its spread, such as lockdowns or physical distancing measures [1]. Lockdowns and social distancing measures interrupted people's lives in many significant ways, leading to social isolation, loss of income, loneliness, inactivity, limited access to basic services, food insecurity, increased alcohol use, online gambling, and decreased family and social support, especially in older and vulnerable people, all potential predictors for a decline in mental health [1,2]. It was therefore not surprising that a hidden mental health crisis was emerging early in the COVID-19 pandemic, both for previously healthy people but especially for those with pre-existing conditions [3]. It was caused by worries about COVID-19 itself, its unpredictability, and associated uncertainties around the fear of infection, fear for loved ones and unknown mortality prospects. Secondary effects of those associated measures might have also led to further mental health issues such as worries about one's children, relationship issues, financial worries, all predictors of poor mental health [4]. Intimate partner violence, highly associated with poor mental health [5], received a lot of attention during the COVID-19 pandemic as lockdowns and restrictions exacerbated risk factors for intimate partner violence while simultaneously reducing access to services for women [6,7].

Evidence has quickly been gathered in high income countries on the mental health burden of COVID-19, yet less is known about the impact of lock downs and restrictions and their secondary effects [8] in low- and middle- income countries, where more than 80% of the world population lives [1,9]. Evidence is even more limited on the impact on women and in resource poor settings where social distancing is difficult to adhere to due to the need to ensure ones livelihoods [10]. Existing evidence on the impact of COVID-19 on societies in low- and middle-income countries has focused on countries that imposed lockdowns or severe restrictions [11,12], and less on countries that only imposed low levels of restrictions or none. Still, it cannot be assumed that COVID-19 had no effect on mental health in those places. The United Republic of Tanzania is one of these countries which had limited restrictions to curb COVID-19 and which were mostly lifted by August 2020 when campaigns for the general election commenced.

The first case of COVID-19 in Tanzania was confirmed on the 16th of March 2020. Subsequently, the government quickly implemented several WHO recommended preventive measures, such as public health information campaigns, closure of schools, banning large gatherings, restricting travel from affected countries, and quarantine of infected people [13]. However, it decided against the implementation of a lockdown to avoid restricting health service access and to allow citizens to continue working to meet their households needs [14]. In June 2020 the Tanzanian government declared Tanzania COVID-19 free leading to a gradual lifting of control restrictions including opening of schools at the end of June 2020 [15]. While people were still cautious and highly aware of COVID-19 in the following months, even though there were no national restrictions, this awareness reduced substantially. By December 2020, there was no visible impact on COVID-19 awareness anymore, for example the wearing of face masks was optional. By the first quarter of 2021 (Jan-March) the situation in the

**Table 1. COVID restrictions timeline in Tanzania.**

| Months/Year | Description of period | COVID-19 control measures/restrictions |
|---|---|---|
| March—June 2020 | Period of strict control measures and intense public awareness campaigns | • The Ministry of Health confirmed the 1st case of COVID-19 on the 16th of March<br>• On 17th of March, all schools closed: nursery, primary & secondary<br>• All unnecessary outdoor gatherings banned including sporting events<br>• Strict hand washing measures introduced<br>• Wearing of face masks mandatory in all government offices<br>• Intense public awareness campaigns |
| July—September 2020 | Period of easing of COVID-19 control restrictions | • Schools opened in July<br>• Wearing of face masks optional & only mandatory in hospitals<br>• Large gatherings permitted<br>• Hand washing measures still in place<br>• Public awareness campaigns continue |
| October—December 2020 | Period of removed restrictions/returning to pre COVID-19 phase | • Public gatherings allowed<br>• Huge public political gatherings for general elections (held on the 28th of October)<br>• Face masks optional & only mandatory in hospitals<br>• Hand washing measures optional |
| January—June 2021 | Period of removed restrictions with life resumed to normal | • Situation returns to pre COVID-19 phase<br>• Face masks optional |

country was similar to the pre-COVID-19 times (see Table 1 for a detailed timeline on the COVID-19 restrictions in Tanzania). This makes Tanzania an ideal case study to investigate the short- and long-term impact of COVID-19 on women's reports of symptoms of poor mental health and the factors influencing this impact.

This paper aims to explore the impact of few and no COVID19 restrictions on the self-reported mental health of women in Mwanza, Tanzania by comparing their reports of symptoms of poor mental health prior, during, shortly after and long after COVID-19 restrictions were implemented through a four-wave longitudinal study.

## Materials and methods

This paper reports on the results of a longitudinal study on COVID-19, that consisted of an existing longitudinal study that incorporated a nested COVID-19 study. That combination resulted in four waves of data collected before (January-March 2020), during (July-September 2020), shortly after COVID-19 restrictions (January-March 2021) and when life has returned to pre-COVID-19 conditions (April-June 2021). The two wave COVID-19 phone survey, conducted between July 2020 and March 2021, was nested within an existing longitudinal face-to-face survey conducted in Mwanza, Tanzania between September 2018 and November 2021. The 415 women recruited into the COVID-19 study were part of the control group of the MAISHA randomized control trial, an empowerment intervention to prevent intimate partner violence among newly formed neighbourhood groups [16]. After completing the endline interview of the trial, women were invited into the longitudinal study. If interested and if they provided informed consent, they were interviewed a third time face-to-face. For the nested

COVID-19 study, interviewers called 455 longitudinal study participants, who had already been interviewed three times through the provided contact details to inform them about the COVID-19 study and recruit them into the nested COVID-19 study. After the two phone interviews for the nested COVID-19 study, one face-to-face interview was also conducted with participants for the original longitudinal study, approximately three months after the last phone interview. Informed consent for the longitudinal study was in written and in person, for the phone interviews, participants provided it via their mobile phones.

The interviewers who were highly experienced in researching sensitive issues were specially trained for three weeks by the principal investigators in conducting phone interviews, ensuring technical readiness of mobile phones of interviewees, such as charged batteries and connection, privacy during the interview to ensure that no one could overhear the conversation and being attuned to dealing with distress or interruption during the interview. The interviewers were also trained to refer women with suicidal thoughts or experiences of violence to available services established for the longitudinal study.

Interviewers scheduled a suitable time for the interviews with the women, ensuring that they would have fully charged and functioning mobile phones and connection. Participants were reimbursed TSH 5,000 for their time through phone money transfer services. Of the 445 eligible women from the longitudinal survey, 433 participated in the first and 426 in the second COVID-19 phone interviewed. Reasons for non-participation were loss to follow up due to moving, being busy and illness. Women were interviewed via phone from July to September 2020 about the impact of COVID-19 in the previous three months and from January to March 2021 about the previous months since restrictions were lifted. In April 2021 women were interviewed a last time face-to-face as part of the overall longitudinal study. The data collected, as displayed in Fig 1, captures the time before COVID-19 occurred for the first time (Wave 1), three months after COVID-19 restrictions were in place (Wave 2), six months after the COVID-19 restrictions were lifted (Wave 3) and three months later, when the pre-COVID-19 life had resumed in Tanzania (Wave 4).

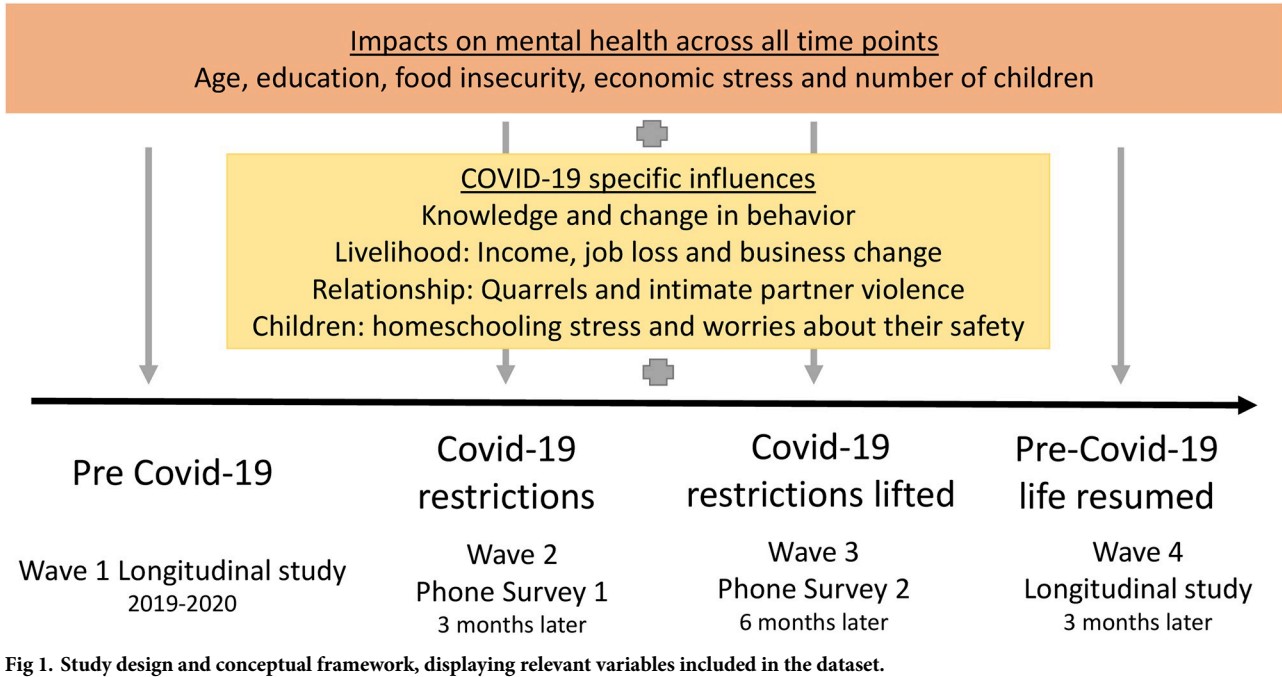

**Fig 1. Study design and conceptual framework, displaying relevant variables included in the dataset.**

The study received ethical approval from the Tanzanian National Health Research Ethics Committee and the London School of Hygiene and Tropical Medicine.

## Measures

Mental health was measured using the Self-Reporting Questionnaire, a 20-item (SRQ20) scale that has been approved by the WHO and validated across many settings, using a cut-off score of 8+ indicating symptoms of poor mental health [17]. As no up to date psychometric properties of the SRQ20 was available for Tanzania, the chosen cut-off point was based on studies conducted in South Africa [18] and Rwanda, with Rwanda being a neighbouring country of Tanzania and in close proximity to Mwanza. It found a good reliability of $\alpha = 0.85$ for the SRQ20 among a random subsample of Rwandan women [19]. Economic stress was assessed by asking if participants were very worried or stressed about their general financial situation. All women who indicated that they have been stressed many times, when responding to the answer options "never, few times or many times" were considered to experience economic stress. Food insecurity was assessed with the question on if they had trouble buying food or other necessities for their families. A response of many times was coded as having experienced food insecurity. We also included the number of children living in the household, coded as zero, one to three and four or more. Age was considered in the categories <30,30–39,40–49, and 50+ and education differentiated between none, primary and secondary education or higher.

In the COVID-19 phone surveys, we asked COVID-19 specific questions, including knowledge of COVID-19. COVID-19 knowledge was assessed through four questions, such as whether treatment is available or whether one can have COVID-19 without showing any symptoms. People with three or more correct questions were considered to have good knowledge of COVID-19. Changes in COVID-19 behaviour included avoiding crowded places, hospitals or going out. If they reported yes to all three of them, they were considered to have changed their behaviour. The cut-off point was determined by the distribution of responses by participants, which showed a high knowledge of COVID-19 and prevention mechanisms in general. In addition to economic stress, impact of COVID-19 on income was measured through questions on whether COVID-19 impacted their income, someone in their household lost a job or whether they had to change their business to ensure income. In terms of relationships, women were asked if during COVID-19 there were more, less or the same number of quarrels in their households. The measurement of intimate partner violence used a shortened version of the WHO Multi-Country Study tool, utilizing four questions based on acts of physical and one question on sexual violence, namely being forced to have sexual intercourse by being threatened, held down or hurt in another way [20]. We also assessed if women reported feeling overwhelmed due to home schooling, whether there was violent disciplining of their children or feeling concerned about their children's safety increased, decreased, or stayed the same during COVID-19. Scales apart from measuring mental health and intimate partner violence were not validated as many questions related directly to the COVID-19 pandemic. Given the timeframe of the study during the early phases of COVID-19, there was no opportunity to test these questions beforehand in addition to the uncertainty around COVID-19 itself. Table 2 therefore displays all variables used in this study, the questions they are based on and how they were coded.

## Analysis

We merged data for the two waves of the COVID-19 dataset with data from the previous and subsequent wave of the longitudinal study yielding a total of four time points (pre COVID-19,

**Table 2. Measures and questions used across the different waves of the study.**

| Measures across all four waves | |
| --- | --- |
| Mental health symptoms | Self-Reporting Questionnaire (SRQ20), using a cut-off point of 8 or more for poor mental health |
| Economic stress | In the past three months were you very worried/stressed about your general financial situation. Would you say, this has happened never, once, few times or many times? Coded many times (Yes) versus few times or never (No). |
| Food insecurity | In the past three months have you had trouble buying food or other necessities for your family? Would you say, this has happened never, once, few times or many times? Coded many times (Yes) versus few times or never (No). |
| Number of children in the household | How many children live in your household? Numeric response was coded as: 0, 1–3 or 4 or more. |
| Age | How old are you in years, coded as: <30,30–39,40–49 and 50+ |
| Education | What is your level of education? Responses coded as: None, Primary education, Secondary education or higher. |
| COVID-19 specific questions in Wave 2 and 3 | |
| Knowledge during COVID-19 | I will read several statements about COVID-19. Please let me know if you agree, disagree, do not know.<br>a. There is currently NO effective treatment for people who already have COVID-19.<br>b. Everyone with COVID-19 will become severely ill at some point.<br>c. It is possible to have COVID-19 without showing any symptoms.<br>d. The virus that causes COVID-19 is spread from one human being to another through respiratory droplets/coughing and sneezing.<br>e. The disease has been eliminated within the country. It no longer exists.<br>Three or more correct questions were coded as good knowledge of COVID-19, 2 or fewer as limited knowledge on Covid-19. |
| Changed behaviour due to COVID-19 | In the previous one month, which, if any, of the following measures have you personally taken to protect yourself or others from the coronavirus (i.e. COVID-19)? *RECORD ALL THAT APPLY (response categories: Yes/No)*<br>a. Avoided going out in general (unless it's important to do so).<br>b. Avoided crowded areas.<br>c. Avoided going to hospital.<br>If they responded yes to all three of them, they were considered to have changed their behavior due to COVID-19. |
| COVID-19 impacted their income | Since the government announced COVID-19, how many times did you not earn enough? Response categories: Never, once, few times, many times, not working at the time being. Coded as yes (Many times) and no (Never, Once, Few times and Not working) to income being sufficient during COVID-19. |
| Job loss due to COVID-19 | Did any of your household members lose their job because of COVID-19? (Yes/No) |
| Change business to ensure income | Have you started any other source of income/business (different from the one you had before) to back you up as a result of COVID-19 (Yes/No/Unemployed) |
| Overwhelmed with home schooling | I felt overwhelmed providing home schooling (Yes/No) |
| Changed violent disciplining of children | Have the children been physically disciplined or hit more, less or the same during the school closure (More/Less/No change) |
| Change in worries about children's safety | Since the school closure in March, do you have more or less concerns about the safety of your children in the house? (More/Less/No change). |
| Changes in household quarrels | Would you say that you have noticed an increase in quarrels in your household since the government announced COVID-19? Or have there been fewer quarrels or they have been about the same? (More/Less/No change) |
| Intimate partner violence | WHO Multicounty Study tool, using last 12 months for Wave 1 and 4, last three months for Wave 2 and 3 to capture physical and /or sexual intimate partner violence. |

during COVID-19 restriction, after COVID-19 restriction were lifted, and after COVID-19 life resumed) for this longitudinal analysis. The analysis was restricted to 415 women who were in a relationship at the time of the survey and participated in all four study waves. Sample characteristics were described using wave 4 dataset. Descriptive analysis was based on calculating frequencies and percentages for categorical demographic variables. We compared COVID-19 knowledge and stressors during COVID-19 and pre-COVID-19 and assessed differences in proportions of participants with good knowledge between surveys using McNemar's Chi-square test of paired proportions for binary variables and Cochran's Q test for variables with more than two categories. Prevalence of symptoms of poor mental health was estimated by wave and the differences assessed using McNemar's Chi-square test, trend analysis was done using random effects model with time as a continuous variable. Three different models were fitted. The first model was a random effects logistic regression model including all four waves' datasets with mental health as an outcome, time, economic stress, food insecurity as random effects exposures. Women age and education were included as time invariant variables. The second model and third models were binary logistic regression models fitted independently for wave 2 (during COVID-19 restrictions) and wave 3 (after COVID-19 restriction were lifted) datasets. All significant COVID-19 knowledge and stressors variables, IPV exposure together with other demographic variables were used as covariates to explain symptoms of poor mental health. In all analyses, a p-value below 0.05 denoted statistical significance.

## Results

The majority of the 415 women in this study, were aged 30 to 49 years old (42%), 40 to 49 years (36%) and 16% were 50 years or older, six percent were aged below 30. More than half of the women had primary education 59% and 25% secondary education or higher. Most women earned money from their own work (81%) and had one to three children (56%), with only three percent having no children and 41% having four or more children. More than 30% of women worried many times about their financial situation, with 14% suffering from food insecurity at least once during the last month.

During the COVID-19 restrictions, half of the women reported that COVID-19 affected their income, 80% changed their business and 29% reported that someone in their household lost a job because of COVID-19. In the time after COVID-19 restrictions were lifted, significantly fewer women reported this as displayed in Table 3. Quarrels in relationships decreased or stayed the same during COVID-19 restrictions, as did intimate partner violence. For intimate partner violence, the trend wasnot significantly different in the months after restrictions though. Women reported feeling overwhelmed providing home schooling to children (27%) and more concerns about their children's safety (61%), which dropped markedly after restrictions ended (8%). There was no report of an increase in violent disciplining of children during COVID-19 restrictions.

Symptoms of poor mental health reported by women peaked at 20% (n = 85) during COVID-19 restrictions compared to a prevalence of 17% (n = 69) before COVID-19 and the 15% (n = 64) after the restrictions were eased and 11% (n = 44) six months later when pre-Covid-19 life resumed (see Fig 2).

The prevalence estimates of mental health symptoms were significantly different between the time during COVID-19 restrictions (Wave 2) and after COVID-19 restrictions were lifted (Wave 3), and after life resumed to pre-COVID-19 times (Wave 4) using McNemar's Exact Chi-square test (p<0.05).

Across all four time points, women were significantly more likely to report symptoms of poor mental health if they were older (OR: 3.9.CI:1.0;15.2 if 40–49 vs <30 and OR:3.8.

**Table 3. COVID-19 Specific knowledge and behaviours after COVID-19 restrictions (n = 415).**

| | | Covid Restrictions | Covid restrictions lifted | P-Value |
|---|---|---|---|---|
| Knowledge about COVID-19 | No | 308 (74%) | 316 (76%) | 0.512* |
| | Yes | 107 (26%) | 99 (24%) | |
| Changed behaviour due to COVID-19 | No | 301 (73%) | 397 (96%) | <0.001* |
| | Yes | 114 (27%) | 18 (4%) | |
| Earn enough during COVID | No | 47 (15%) | 98 (28%) | <0.001* |
| | Yes | 274 (85%) | 246 (72%) | |
| Household member lost a job | No | 293 (71%) | 385 (93%) | <0.001* |
| | Yes | 122 (29%) | 30 (7%) | |
| Other source of income | No | 330 (80%) | 366 (88%) | <0.001** |
| | Yes | 65 (16%) | 30 (7%) | |
| | Unemployed | 20 (5%) | 19 (5%) | |
| Overwhelmed with home schooling | No | 285 (73%) | 345 (87%) | <0.001* |
| | Yes | 104 (27%) | 52 (13%) | |
| Changed violent disciplining of children | No change | 186 (46%) | 189 (46%) | 0.875** |
| | Less | 198 (49%) | 216 (53%) | |
| | More | 22 (5%) | 6 (1%) | |
| Change in worries about children safety | No change | 42 (10%) | 189 (46%) | <0.001** |
| | Less | 117 (29%) | 189 (46%) | |
| | More | 247 (61%) | 33 (8%) | |
| Change in household quarrels | No change | 270 (65%) | 269 (65%) | 0.931 |
| | Less | 113 (27%) | 130 (31%) | |
| | More | 32 (8%) | 16 (4%) | |

* McNemars Chi Square test,

** Cochran's Q Chi Square test.

CI:0.9;15.5), had low levels of education (OR: 0.44. CI:0.9;15.5 if secondary or higher versus none), experienced food insecurity (OR:3.7. CI:2.4;5.9) and were worried about their financial situation (OR: 2.9. CI:1.9;4.5).

The univariate analysis results in Table 4 show that the key factors associated with symptoms of poor mental health during COVID-19 restrictions included a change in behaviour due to COVID-19, feeling overwhelmed by having to provide home schooling, increased quarrels in the household, physical and/or sexual violence perpetrated by the partner and not having secondary education. In the multivariable model, all factors remained significant, even after controlling for each other. After restrictions were lifted, symptoms of poor mental health were associated with being more concerned about children's safety, feeling overwhelmed providing home-schooling, increased or fewer quarrels in the home, the occurrence of physical and/or sexual intimate partner violence and women lower level of education. All factors remained significant in the multivariate model controlling for those factors that were also significant in the univariate analysis.

## Discussion

To our knowledge, this is one of the first studies to investigate the prevalence of symptoms of poor mental health longitudinally before, during and after COVID-19 in a setting with limited COVID-19 restrictions imposed by the government but a strong initial public health information campaign. Despite Tanzania imposing only low levels of restrictions to control the

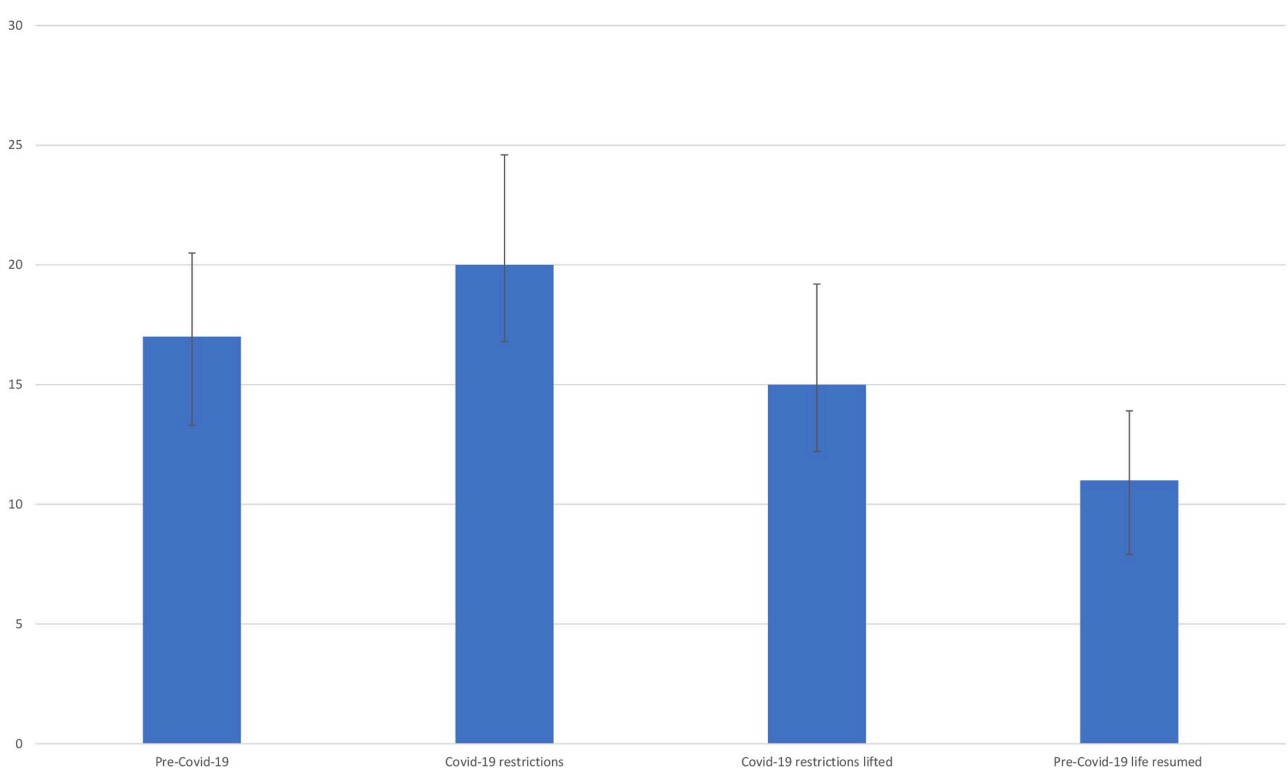

**Fig 2. Prevalence of mental health symptoms by wave [11].**

pandemic, we have still seen in this study that COVID-19 and its restrictions led to an increase in women's reports of symptoms of poor mental health from 17% before COVID-19 to 20% during COVID-19 restrictions, which significantly decreased a few months after the restrictions were lifted (15%) and fell even further once life resumed to pre-COVID-19 times (11%). The pre-COVID-19 prevalence of symptoms of poor mental health of 17% is comparable to that found internationally of 17.6% [21] and the prevalence during COVID-19 of 20% is markedly lower than that of 39% established for Africa through a systematic review [22].

Symptoms of poor mental health throughout the period were explained through the key socio-demographic indicators such as women's age, food insecurity and economic stress. These factors were exaggerated during the pandemic by COVID-19 specific stressors such as behaviour change, livelihood factors, such as losing a job and insufficient income [23,24]. Relationship factors such as intimate partner violence and increased quarrels, as well as children related stress due to home schooling and concerns about the safety of children also played a part [25]. These factors still impacted women's reporting of poor mental health symptoms after COVID-19 restrictions were lifted, although all of them were less frequent. Intimate partner violence has long been established as a key cause of women's poor mental health [5,26,27], and numerous studies worldwide showed that it increased globally during the COVID-19 pandemic [28]. While our study could not investigate a change in intimate partner violence before and after the COVID-19 pandemic due to different time frames of the measurement, no difference was observed between the time of restrictions and after restrictions were lifted.

Concerns regarding livelihood challenges due to COVID-19 had a significant impact on women's reporting of symptoms of poor mental health, which resonates with studies across the globe that link economic stress to increases in overall stress and the need to look for work

**Table 4. Covid-19 specific factors associated with symptoms of poor mental health during COVID-19 restrictions.**

| | | COVID-19 restrictions | | | | Covid-19 no restrictions | | | |
|---|---|---|---|---|---|---|---|---|---|
| | | Total | Symptoms of poor mental health | Odds ratio | 95% confidence interval | Total | Symptoms of poor mental health | Odds ratio | 95% confidence Interval |
| | | N = 415 | N = 85 | | | N = 415 | N = 64 | | |
| Age | <30 | 24 (6%) | 2 (2%) | 1.00 | | 24 (6%) | 1 (2%) | 1.00 | |
| | 30–39 | 174 (42%) | 33 (39%) | 2.57 | (0.58;11.5) | 174 (42%) | 24 (38%) | 3.68 | (0.47;28.5) |
| | 40–49 | 149 (36%) | 33 (39%) | 3.13 | (070;14.0) | 149 (36%) | 26 (41%) | 4.86 | (0.63;37.6) |
| | 50+ | 68 (16%) | 17(20) | 3.67 | (0.78;17.2) | 68 (16%) | 13 (20%) | 5.44 | (0.67;44.0) |
| Education | None | 67 (16%) | 19(22%) | 1.00 | | 67 (16%) | 17 (27%) | 1.00 | |
| | Primary | 244 (59%) | 53(62%) | 0.70 | (0.38;1.29) | 244 (59%) | 37 (58%) | 0.53 | (0.27;1.01) |
| | Secondary/ higher | 104 (25%) | 13(15%) | 0.36 | (0.16;0.79) | 104 (25%) | 10 (16%) | 0.31 | (0.13;0.73) |
| COVID-19 knowledge | No | 308 (74%) | 61 (72%) | | | 316 (76%) | 49 (77%) | | |
| | Yes | 107 (26%) | 24 (28%) | 1.17 | (0.69;2.00) | 99 (24%) | 15 (23%) | 0.97 | (0.52;1.82) |
| COVID-19 behavior change | No | 301 (73%) | 53 (62%)* | 1.00 | | 397 (96%) | 60 (94%) | 1.00 | |
| | Yes | 114 (27%) | 32 (38%) | 1.83 | (1.10;3.03) | 18 (4%) | 4 (6%) | 1.60 | (0.51;5.04) |
| Income sufficient during COVID-19 | No | 160 (50%) | 16 (23%)*** | 1.00 | | 260 (76%) | 28 (51%)* | 1.00 | |
| | Yes | 161 (50%) | 53 (77%) | 2.56 | (0.97;6.74) | 84 (24%) | 27 (49%) | 1.72 | (0.85;3.49) |
| Business changes due to COVID-19 | Yes | 65 (16%) | 17 (20%) | 1.00 | | 30 (7%) | 8 (13%) | 1.00 | |
| | No | 330 (80%) | 64 (75%) | 1.47 | (0.79;2.73) | 366 (88%) | 54 (84%) | 2.10 | (0.89;4.96) |
| | Not working | 20 (5%) | 4 (5%) | 1.04 | (0.34;3.21) | 19 (5%) | 2 (3%) | 0.68 | (0.15;3.03) |
| Job loss due to COVID-19 | Yes | 122 (29%) | 33 (39%)* | 1.00 | | 30 (7%) | 7 (11%) | 1.00 | |
| | No | 293 (71%) | 52 (61%) | 1.72 | (1.04;2.83) | 385 (93%) | 57 (89%) | 1.75 | (0.72;4.27) |
| Quarrels due to COVID-19 | Same | 270 (66%) | 41 (9%)*** | 1.00 | | 269 (65%) | 28 (44%) | 1.00 | |
| | Fewer | 113 (27%) | 28 (33%) | 1.84 | (1.07;3.16) | 130 (31%) | 28 (44%) | 2.36 | (1.33;4.19) |
| | More | 32 (8%) | 16 (19%) | 5.59 | (2.59;12.1) | 16 (4%) | 8 (13%)*** | 8.61 | (3.00;24.7) |
| Overwhelmed providing home schooling | Yes | 104 (27%) | 32 (40%)** | 1.00 | | 52 (13%) | 16 (27%)*** | 1.00 | |
| | No | 285 (73%) | 48 (60%) | 2.19 | (1.31;3.69) | 345 (87%) | 43 (73%) | 3.12 | (1.60;6.10) |
| Violent disciplining due to COVID-19 | Same | 186 (46%) | 32 (39%) | 1.00 | | 189 (46%) | 27 (44%) | 1.00 | |
| | Less | 198 (49%) | 41 (49%) | 1.26 | (0.75;2.10) | 216 (53%) | 32 (52%) | 1.04 | (0.60;1.82) |
| | More | 22 (5%) | 10 (12%)** | 4.01 | (1.60;10.1) | 6 (1%) | 2 (3%) | 3.00 | (0.52;17.2) |

(*Continued*)

**Table 4.** (Continued)

| | | COVID-19 restrictions | | | | Covid-19 no restrictions | | | |
|---|---|---|---|---|---|---|---|---|---|
| | | Total | Symptoms of poor mental health | Odds ratio | 95% confidence interval | Total | Symptoms of poor mental health | Odds ratio | 95% confidence Interval |
| | | N = 415 | N = 85 | | | N = 415 | N = 64 | | |
| *Concerns for children's safety due to COVID-19* | Same | 42 (10%) | 7 (8%)*** | 1.00 | | 189 (46%) | 22 (36%)** | 1.00 | |
| | Less | 117 (29%) | 10 (12%) | 0.47 | (0.17;1.32) | 189 (46%) | 28 (46%) | 1.32 | (0.73;2.40) |
| | More | 247 (61%) | 66 (80%) | 1.82 | (0.77;4.30) | 33 (8%) | 11 (18%) | 3.80 | (1.62;8.87) |
| *Sexual and/or physical intimate partner violence* | No | 386 (93%) | 72 (85%)*** | 1.00 | | 384 (93%) | 51 (80%) | 1.00 | |
| | Yes | 29(7%) | 13 (15%) | 3.54 | (1.63;7.69) | 31 (7%) | 13 (20%)*** | 4.72 | (2.18;10.2) |

***significant at p<0.001.

**significant at p-value<0.005.

*significant at p-value<0.05.

outside the home despite the fear of infection [29,30]. Similarly, a study in the United States of America found that parents who experienced stress, for example due to home schooling or concerns about their children's safety, also experienced higher levels of anxiety and depressive symptoms [31]. While this study reported a high association with child maltreatment and parental stress [31], women in our study reported less violent discipling of their children than before COVID-19. One reason for the high child maltreatment observed in the American study could be that the women and their partners spent more time with their children at home during lockdowns.

There are several limitations that need to be considered in this study. The study used phones to interview women, a method that has not previously been tested with this population and that might have led to underreporting of symptoms of poor mental health, experiences of intimate partner violence and violent disciplining of children. As the study aimed to examine the impact of COVID-19 restrictions in Tanzania, the time frames of the intimate partner violence measurement of three months were not comparable to the time frame of 12 months used in the pre-COVID-19 waves. Furthermore, the sample size is comparatively small with 415 women, as we could only recruit women who previously consented to participate in the longitudinal study and provided their contact details. In order to keep the interview short, no questions were asked on potentially important factors associated with symptoms of poor mental health such as alcohol use, emotional or economic abuse, and inability to buy nutritious food or illness. In particular, as diagnostics for COVID-19 were still absent at this time, it was not possible to assess whether participants or their family members suffered from COVID-19 or had particular health conditions that made them more vulnerable.

Overall, this study showed that low levels of restrictions as imposed by the government of Tanzania during the COVID-19 pandemic led to an increase in symptoms of poor mental health, albeit not as pronounced as in settings with strict restrictions or lockdown [1]. This has important implications for decision making in future pandemics when governments need to weigh the potential harm of implementing control measures to contain infectious diseases and yet protect the population from short and long term harm from non-communicable diseases such as mental health. Even if no or low levels of restrictions are chosen, adequate support needs to be given to the population to avoid increased anxiety and challenges to economic

livelihoods [10]. In particular, attention needs to be given to the double burden that women face in respect to reduced income generating activities, relationship pressures and increased childcaring responsibilities.

## Conclusion

The negative effects of pandemic outbreaks such as COVID-19 have far-reaching consequences to vulnerable populations in low- and middle-income countries with poor mental health services. The results from this study and others in Africa [20] emphasize the need to ensure that the far-reaching mental health consequences of disease outbreaks and the containment measures are closely monitored and addressed. Setting up or strengthening existing community based mental health services, with specific focus on vulnerable groups such as women and children should be part of national and international preparedness efforts.

## Acknowledgments

We want to thank all our participants for their time and contributions. Special thanks also to the survey team for their hard work and dedication.

## Author Contributions

**Conceptualization:** Heidi Stöckl, Elizabeth Dartnall, Grace Mtolela, Gerry Mshana.

**Data curation:** Neema Mosha, Philip Ayieko, Grace Mtolela, Gerry Mshana.

**Formal analysis:** Heidi Stöckl, Neema Mosha, Philip Ayieko, Gerry Mshana.

**Funding acquisition:** Heidi Stöckl, Gerry Mshana.

**Investigation:** Heidi Stöckl, Elizabeth Dartnall, Philip Ayieko, Grace Mtolela, Gerry Mshana.

**Methodology:** Heidi Stöckl, Philip Ayieko, Gerry Mshana.

**Project administration:** Grace Mtolela.

**Software:** Heidi Stöckl, Neema Mosha.

**Supervision:** Gerry Mshana.

**Validation:** Heidi Stöckl.

**Writing – original draft:** Heidi Stöckl.

**Writing – review & editing:** Heidi Stöckl, Neema Mosha, Elizabeth Dartnall, Philip Ayieko, Grace Mtolela, Gerry Mshana.

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
