## [Decision Letter · Decision Letter 0]

19 Jan 2023

PGPH-D-22-02046

The short and long-term impact of COVID-19 restrictions on women’s mental health symptoms in Mwanza, Tanzania: a longitudinal study.

Dear Dr. Stöckl, PhD,

Thank you for submitting your manuscript to PLOS Global Public Health. After careful consideration, we feel that it has merit but does not fully meet PLOS Global Public Health’s publication criteria as it currently stands. Therefore, we invite you to submit a revised version of the manuscript that addresses the points raised by the reviewers during the review process.

We look forward to receiving your revised manuscript.

Kind regards,

Rakesh Singh

Academic Editor

Journal Requirements:

1. Please provide separate figure files in .tif or .eps format only and remove any figures embedded in your manuscript file. Please also ensure that all files are under our size limit of 10MB.

2. In the online submission form, you indicated that "Data is available from the authors upon request". All PLOS journals now require all data underlying the findings described in their manuscript to be freely available to other researchers, either 1. In a public repository, 2. Within the manuscript itself, or 3. Uploaded as supplementary information.

Additional Editor Comments: Kindly ensure you follow the journal requirements while revising your manuscript. 

It'd suggest the authors to add relevant literature in the introduction and discussion to strengthen the sections from the following papers. 10.1016/j.ajp.2020.102259, 10.1016/j.ajp.2021.102776, 10.1016/j.ajp.2020.102222, 10.52095/gp.2021.3838.1027

Reviewers' comments:

Reviewer #1: Title:

The word “symptoms” seems extra, should be removed.

Abstract:

The abstract does not report the objective and findings of the study clearly. Sentences are quite vague to be understood easily.

Introduction:

Introduction does not include the rationale, significance, and scope of the study.

Method:

The psychometric properties of SRQ20 are not presented. Method fails to highlight the objective and the design of the study.

Results:

Tables are not in appropriate format e.g. APA.

Discussion:

Discussion fails to signify the findings of the study.

Reviewer #2: 1. While the manuscript adhere to the PLOS standards of publication, there are issues in certain sections relating to methods and measures sections that are included in the report.

2. No issues with the statistical part.

3. Authors have mentioned that the data will be available upon request.

4. The writing of some sections need to be more clear. This is indicated in the report.

---

## [Decision Letter · Decision Letter 1]

22 May 2023

The short and long-term impact of COVID-19 restrictions on women’s mental health in Mwanza, Tanzania: a longitudinal study.

PGPH-D-22-02046R1

Dear Dr Stöckl, PhD,

We are pleased to inform you that your manuscript 'The short and long-term impact of COVID-19 restrictions on women’s mental health in Mwanza, Tanzania: a longitudinal study' has been provisionally accepted for publication in PLOS Global Public Health.

Best regards,

Rakesh Singh

Academic Editor
